# Clinically Applicable Assessment of Tisagenlecleucel CAR T Cell Treatment by Digital Droplet PCR for Copy Number Variant Assessment

**DOI:** 10.3390/ijms23147573

**Published:** 2022-07-08

**Authors:** Soragia Athina Gkazi, Emma Gravett, Carla Bautista, Jack Bartram, Sara Ghorashian, Stuart Paul Adams

**Affiliations:** 1SIHMDS-Haematology, Great Ormond Street Hospital for Children NHS Foundation Trust, London WC1N 3HJ, UK; emma.gravett@gosh.nhs.uk (E.G.); c.bautista@ucl.ac.uk (C.B.); 2Department of Haematology, Great Ormond Street Hospital for Children NHS Foundation Trust, London WC1N 3HJ, UK; jack.bartram@gosh.nhs.uk (J.B.); sara.ghorashian@gosh.nhs.uk (S.G.)

**Keywords:** ddPCR, tisagenlecleucel, 4-1BB CD19, copy number variant

## Abstract

Chimeric antigen receptor (CAR) T cell therapy is an innovative immunotherapy for treating cancers in both children and adults with proven utility in numerous clinical trials. Significantly, some CAR T cell therapies have now been approved by relevant national regulatory bodies across numerous countries for clinical therapeutic use outside of clinical trials. One such recently licensed product is tisagenlecleucel, a CAR T therapy approved for the treatment of B-cell acute lymphoblastic leukemia (B-ALL) using autologous T cells from the patient. The genetically engineered T cells target a protein called CD19, common to B cells, through a CAR incorporating a 4-1BB costimulatory domain to improve response. Since tisagenlecleucel is now a standard of care treatment for B-ALL, it is clinically essential to be able to accurately monitor these CAR T cells in patients. Assessment of the copy number variant (CNV) of the CAR T cell products allows this within a clinically acceptable timeframe for optimal patient benefit. However, no standardized method with high reproducibility and efficiency has been described within a routine clinical laboratory setting. Here, we demonstrated a novel digital droplet PCR (ddPCR)-based methodology for the study of CNV (ddPCR-CNV) in 4-1BB CD19-specific CAR T cells with universal applicability across clinical diagnostic laboratories.

## 1. Introduction

The development of chimeric antigen receptor T cell (CAR T) therapy has permitted novel intervention for previously untreatable cancers and has been used in many clinical trials to treat various forms of cancer across many countries. CD19-targeted chimeric antigen receptor cell therapy shows great promise for relapse or refractory pediatric B cell acute lymphoblastic leukemia (B-ALL) [1,2,3]. Typically, CAR T cells targeting CD19 utilize either a 4-1BB or a CD28 signaling domain, expressed using lentiviral vectors [4], to enhance the antileukemic effect. While early interventions with CD19-directed CAR T therapy were used purely on a clinical trial basis, it is now possible to use licensed products in many centers. Tisagenlecleucel is a commercially available autologous second generation CD19 CAR T therapy utilizing a 4-1BB signaling domain expressed using lentiviral transduction [4] (Novartis) that has been approved for use by the FDA in the United States and the EMA in Europe, leading to widespread access. 

Following CD19 CAR T therapy, initial response rates for relapse or refractory pediatric B-ALL are 80–90%, but ultimately approximately 50% of patients will relapse because of either loss of CD19 expression on the tumor cells [5] or a lack of CAR T cell persistence [6,7]. To date, centers have used the association of loss of B cell aplasia as a correlate of lack of CAR T cell persistence, but B cell aplasia has no standard definition and can lag behind loss of CAR T cell persistence by many months, especially if restricted to assessment of peripheral blood B cell populations. In the intervening period, relapse can occur. Assessment of B cell populations in the bone marrow (BM) can provide earlier evidence of loss of CAR T cell persistence, but this approach is not standardized and requires a clinical procedure (bone marrow aspiration—BMA) to be performed, with logistical and clinical risk ramifications. As a result, this limits the frequency of testing. Given the widespread use of CD19 CAR T therapy in both clinical trials and the clinic and the association of lack of CAR T persistence with relapse in ALL [8,9], real-time assessment of CAR T cell persistence is imperative in routine diagnostic laboratories. Assessment by copy number variation (CNV) permits measurement of CAR T numbers and persistence in patients at sequential timepoints to assist subsequent clinical decision making, including on the need for further therapy, e.g., stem cell transplantation, should early loss of CAR T cell therapy occur [10,11,12]. To date, the detection and quantification of CAR T cells, both immediately after treatment and as a follow-up, has remained a technical challenge [13,14,15,16,17,18,19,20,21]. Here, we report the development of a protocol for the detection of CNV specifically for the tisagenlecleucel 4-1BB signaling domain (4-1BB-CAR). To meet the needs for a robust and rapid protocol, a ddPCR-based technique was developed (ddPCR-CNV) that showed more consistent and lower limit of detection than previously published techniques [7,10,11,22]. In this report, we demonstrate this methodology to be efficient and accurate for the monitoring of CAR T cells, especially in pediatrics. Crucially, it is sensitive enough for effective monitoring in peripheral blood samples, potentially negating the need for frequent, invasive BMA. Importantly, it is standardized and ready for implementation, with the possibility of reproducibility of the protocol in other CAR T cell treatments and not only CD19. 

## 2. Results

To determine the limit of detection (LoD), we ran five (A–E) nonamplifiable controls (NACs) in triplicate. These were tisagenlecleucel-negative and reference gene-positive so that CNV could be calculated for the target. The mean CNV for the controls was 0.001, with a range of 0.000–0.003, demonstrating the limits between CNV detection and background noise. Therefore, the LoD for the CNV of tisagenlecleucel was set to be 0.001 (Figure 1A). The synthetic oligonucleotide was diluted at different absolute copy numbers (with no presence of genomic DNA) from 10^6^ down to 0.0001 copies/µL, including a nontemplate control (NTC) (Figure 1B). This experiment was performed to (i) determine the dynamic range of the ddPCR (previously suggested by the manufacturer, Bio-Rad, to be 1–10^5^), (ii) validate correlation between the observed final concentrations of the samples and the expected values, and (iii) establish the appropriate level of target dilution to use for spiking. The results showed a positive trend between observed and expected values between 1 and 10^5^ copies/µL, demonstrating saturation above 10^5^ and loss of resolution below 1 copy per µL, as expected (Figure 1B). Therefore, 10^4^ copies/µL was established as the most appropriate dilution to be used in the following experiments. This amount demonstrated a good balance and distinction between positive and negative droplets, allowing for a more accurate CNV measurement (Figure 1C). An NTC was used throughout all runs, showing a limit of blank (LoB).

The 4-1BB-CAR T cell frequency in patient samples varied depending on the disease level prior to tisagenlecleucel administration, date of sample collection, and sample condition. Low target frequencies were expected, making it difficult to precisely predict the CNV patterns per sample. Consequently, the potential CNV outcome of 4-1BB-CAR T reactions was examined in the first step by limit of quantification (LoQ) based on each patient’s sample taken prior to CAR T administration. As demonstrated in Figure 2, each sample (S1, S2, S3, and NTC) was run in triplicate to validate the consistency of the protocol. In this case, two of the samples were found to be above their LoQs (S2 and S3), demonstrating CAR T persistence, while S1 was just below, suggesting a potential CAR T loss.

To validate the efficacy and consistency of the protocol, we used high levels of the 10^4^ oligo spiked in NACs in order to be able to easily detect it. Three different genomic DNA control samples (NAC1, NAC2, NAC3) were used in triplicate to determine which one showed the most consistent data to use in the following step (Figure 3A). A one-way ANOVA test showed no significant differences among the three samples (*p =* 0.998539). Both the synthesized 4-1BB-CAR target oligo and the chosen NAC1 control genomic DNA were diluted down to 2000 copies/µL. Then, a series of spikes of the oligo in the control genomic DNA were performed to determine final concentration in copies/µL. The final concentrations were found to follow a linear trend with the percentages of spikes (R^2^ = 0.9845) (Figure 3B). In Figure 3C, the interchange between target concentration and reference concentration, from 100% to 0%, is visualized. The controls spiked with different percentages of the tisagenlecleucel oligo were also examined to set the threshold range for tisagenlecleucel probes during analysis (Figure 3D,E). Analysis was performed while setting the FAM threshold from 500 up to 3000, showing the ideal range to be between 600 and 2500. Figure 3D shows the consistency of the results lost below 600 and above 2500. Representative results of three points, with thresholds set at 500, 1500, and 3000 are shown as 2D plots in Figure 3E. The results were consistent between runs. Representative data from target spiking in NACs at realistic copies showed detectable events at as low as 0.001 copies/µL (Figure 3F and Table A1).

An example of the outcome of ddPCR-CNV and data analysis is shown in Figure 4. A setting of the appropriate threshold in 2D amplitude so that the four quadrants are distinct is shown in Figure 4A. The results from a representative ddPCR-CNV experiment using longitudinal samples from two different patients are shown in Figure 4B,C, depicting the power of this protocol to identify tisagenlecleucel copies as early as 7 days posttreatment (Figure 4B) and at very low levels. Raw data can be found in Appendix E.

One strength of this protocol is the limited sampling needed from pediatric patients. Differing DNA inputs were assessed by running DNA from (1) 0.1 ng/µL up to 200 ng/µL (Figure 5A), (2) isolated CD3+ T cells and PBMNC (Figure 5B), and (3) PBMNC and BMA (Figure 5C) from the same patient and at the same timepoint. Interestingly, CNV measurements did not change when using as little as 10 ng/µL DNA from bulk PBMNC, removing the need for a BMA or a CD3+ T cell isolation. Finally, reporting CNV numbers from either PBMNC or BMA from the same patient timepoint did not affect the outcome of the clinical report.

## 3. Discussion

Digital droplet PCR is a quantitative PCR (qPCR) assay that is partitioned into multiple single reactions (droplets), comparably to a limiting dilution approach. Compartmentalization of the PCR mix can be achieved in many different ways. However, here, a droplet-based technique in which a PCR reaction mix was divided into ~20,000 droplets was used. The limits of quantification (LoQs) and limits of detection (LoDs) were quantified, and a comprehensive methodology for tisagenlecleucel quantification was developed. The specific primer design can be adapted for other CAR T cell therapies, permitting the assay to be used for any indication and CAR T cell product based on selection of a transgene-specific primer set.

Although other assays have been used for the detection and quantification of CAR T cell persistence, notably flow cytometry and qPCR, ddPCR has some significant advantages:(i)Flow cytometry assays work similarly to ddPCR to detect and quantify the fluorescence signal in the stained test set (CD19 for tisagenlecleucel). Here, cells are used directly. However, sample handling and processing, as well as assay times, can prove time consuming, and the results are less consistent than those of molecular assays. Flow cytometry techniques are operator dependent and difficult to standardize, and they have been proven before to be quantitatively less accurate than molecular assays [5,23].(ii)qPCR is fast and inexpensive. It is a well suited technique for the quantification of CAR T cell CNV. However, standards are needed for qPCR, and its sensitivity has been shown to be lower than that of ddPCR. In addition, more starting material is needed for qPCR [24].(iii)ddPCR-CNV offers a quick, reliable method for the detection of CAR T cell persistence. Similar assays have been described both theoretically and methodologically elsewhere [15,20], demonstrating that ddPCR is often more robust than qPCR while requiring lower input of DNA. ddPCR-CNV requires access to specialized equipment. The tool combines high sensitivity with excellent accuracy; it is comparatively fast (results are available within <3 days from sample collection and receival in the lab, Appendix D) and inexpensive (approximately $3.6 per sample). Thus, it represents an excellent alternative to the existing methods for patient disease monitoring and can provide real-time assessment of disease in patients, which will significantly benefit clinical teams with treatment options and clinical management of the patient.

This ddPCR-CNV assay is valuable for quantifying and monitoring tisagenlecleucel CAR T persistence and can easily be adapted for use with other CAR T cell emerging therapies. Importantly, this ddPCR-CNV does not require standards, making it very useful for CNV quantification in peripheral blood, bone marrow, and biopsies. Additionally, only 20 ng DNA (from bulk PBMNC) per reaction is used in this protocol (Figure 5A), significantly reducing required input DNA/cell number per test in comparison with other techniques. This is of particular importance in pediatric patients because of the low sample volumes feasible at any given timepoint. 

There are pivotal controls required for this ddPCR assay: a no-template control (NTC) and the previously ddPCR-CNV-determined patient samples. The NTC should include all components for the reaction except for the gDNA, enabling exclusion of contamination. Consequently, the NTC should show no PCR signals in the final analysis. The previously ddPCR-CNV-determined patient samples serve as positive controls. In the final analysis, the CNV values for the positive controls should be consistent between runs to ensure accuracy and reproducibility (Figure 2). All samples should be run in triplicate because of the expected low CNV frequencies. Where possible, the LoQ should be calculated on a sample taken from the same patient prior to treatment (Figure 4B). It is therefore essential to set a consistent range of thresholds to distinguish between positive and negative events. For precise analysis, populations of negative and positive events must be easily discriminated (Figure 4A), and therefore, PCR parameter adjustments may be needed (Figure 3D,E). When the number of PCR cycles increases, the fluorescence intensities of positive events also increase. Adjusting the melting temperature (Tm) during qPCR is another way to improve positive and negative event distinction. If the positive and negative signals are distinguishable, threshold setting adjustment should have negligible effect on CNV calculations.

To summarize, a novel ddPCR approach for tisagenlecleucel CAR T cell detection in patient samples is presented, describing the LoD and LoQ of the test, the controls required, and a step-by-step guide suitable for use in a routine clinical laboratory setting (Appendix D). General ddPCR advice is also provided for this method to be adapted for accurate quantification of other CAR T cells, much needed in this rapidly advancing field. This level of “real-time” monitoring in essential for this important group of patients in order to predict outcomes and plan interventions in a timely fashion. The power of the technique developed herein stems from the limited sample needed (crucial for patients from whom only small volumes can be taken) combined with the limits of detection and quantification alongside the high accuracy of the results provided. In parallel to previous studies based on the development of a ddPCR technique to monitor CAR T cells [13,14,15,17,18,20,21], this paper emphasizes the importance of the technique, especially for pediatric patients, for whom sampling is usually limited. Moreover, a PBMNC sample can be used without loss of sensitivity, and we showed here that a BMA sample was not necessary for the accurate measurement of CNV (Figure 5C) and that there was no need for CD3+ T cell isolation (Figure 5B) for the tisagenlecleucel target, reducing sample processing times and costs. The methodology described here has now been successfully used in more than 150 samples from children, and a schematic of the procedure can be seen in Figure 6.

## 4. Materials and Methods

An overview of the method is provided in Figure 6.

### 4.1. Primer, Probe, and Synthetic Oligonucleotide Design (Steps 1–3)

The ddPCR-CNV described here was based on the measurement of fluorescence upon binding and hydrolysis of the FAM probe specifically binding to the specified target region. ZEN probes are internal quenchers that shorten the distance between the 5′ dye (FAM) and quencher (traditionally this distance is 20–30 bases), and in concert with the 3′ nonfluorescent quencher (NFQ), they provide greater overall dye quenching, lowered background, and increased signal detection. With their higher annealing temperatures, ZEN probes bind to their DNA strands earlier than the PCR primers; thus, the probes are already bound when the elongation phase of the PCR begins. Upon DNA polymerization, the ZEN probe is cleaved, releasing the 5′ fluorophore from being controlled by the quencher; FAM, in this instance, becomes detectable. If a given probe cannot bind to its amplicon, no fluorescence signal is be emitted. As the synthetic oligo was only 62 bp long, the primers and probe specific to it had to be designed consecutively (but not overlapping), with the probe being in the middle of the two primers. Details regarding the reagents and consumables can be found in Table A1 and Table A2.

STEP 1: Design of the 4-1BB-CAR T-specific primers followed these guidelines:(i)The size of the PCR product should ideally be <100 bp (the oligo here was 62 bp). Note that the shorter the product, the higher the efficiency of the technique (Bio-R).(ii)The GC content of the PCR product should ideally be 40–60%.(iii)Secondary structures should be avoided.(iv)Primer length should be 18–25 nucleotides, with a GC content of 50–60%.(v)Primers should not contain long stretches of Gs or Cs (longer than 3 bases).(vi)If possible, Gs and Cs should be placed at the ends of the primers (but no more than a stretch of 2).(vii)Primer annealing temperature should be ~60 °C.(viii)To exclude the formation of primer dimers, 3′ complementarity of forward and reverse primers should be avoided.

STEP 2: The probe was designed according to the following guidelines:(i)The probe must be located between the two primers.(ii)The probe sequence cannot overlap with either primer but can be directly adjacent to either or both.(iii)The probe should be shorter than 30 nucleotides, as it has been shown that longer distances between the fluorophore and quencher can increase background signal intensity.(iv)Tm (probe) should be ~5 degrees higher than those of the primers; i.e., Tm (probe) should be ~65 °C if the Tms of the primers are ~60 °C. If this is impossible, a range of 3–10 °C is still considered acceptable.(v)GC content should be 40–60 °C.(vi)Probes should not have a G at the 3′ or 5′ ends.(vii)Probe labelling: as ddPCR-CNV is based on a duplex assay, the probes should be labelled with either FAM or HEX. Alternatively to HEX, VIC can be used. A HEX and VIC probe should not be combined in the same assay.(viii)Nonfluorescent quenchers (e.g., BHQ or DDQ) should be used. Quenchers are essential for blocking the fluorescence signal, and they are usually at the 3′ end, whereas the fluorescent label is at the opposite side, the 5′ end. These need to be chosen during the ordering process. MGB and LNA probes are also options; they are usually shorter than standard ones.

The following primers and probe were specifically designed for detection of the 4-1BB-CAR T cells and the synthetic construct (all supplied by Integrated DNA Technology, Iowa, US): forward for tisagenlecleucel (TGCCGATTTCCAGAAGAAGAAGAAG), reverse for tisagenlecleucel (GCGCTCCTGCTGAACTTC), probe tisagenlecleucel (ACTCTCAGTTCACATCCTC). The tisagenlecleucel probe was labelled with a FAM dye and a nonfluorescent quencher (ZEN). In addition, the following primers and probe were designed for control gene (albumin) detection: forward for albumin (TGAAACATACGTTCCCAAAGAGTTT), reverse for albumin (CTCTCCTTCTCAGAAAGTGTGCATAT), probe albumin (TGCTGAAACATTCACCTTCCATGCAGA). The albumin probe was labelled with a HEX dye and nonfluorescent quencher (ZEN). 

STEP 3: A 62 bp synthetic oligonucleotide, based on the 4-1BB zeta domain of the CD19 and specific for the genetically engineered cells, was designed to determine the LoD of the CD19 CAR T reactions (TGCCGATTTCCAGAAGAAGAAGAAGACTCTCAGTTCACATCCTCGAAGTTCAGCAGGAGCGC). 

### 4.2. Sample Preparation (Steps 4–5)

Peripheral blood or bone marrow was collected into EDTA tubes from patients receiving tisagenlecleucel CAR T cell therapy as part of their standard of care for routine clinical testing. Isolation of the mononuclear cells (MNCs) was performed using a density gradient with centrifugation [15] (STEP 4), and DNA was extracted from the MNCs using the QIAamp DNA Blood Mini Kit as described previously [18,25,26] (Qiagen, Hilden, Germany). Extracted DNA was quantified using the Nanodrop 1000 (Labtech Ltd., East Sussex, UK) and diluted to 10 ng/µL (STEP 5). When CD3 T cells were isolated, Miltenyi Biotec microbead kits were used.

### 4.3. Digital Droplet PCR (DDPCR)—Steps 6–9

For general ddPCR, a reaction mixture was set up using a PCR mastermix and then partitioned into droplets using a droplet generator. We used the Automatic Droplet Generator (Bio-Rad Laboratories Inc., Watford, UK), which generated the droplets directly into a 96-well plate. The plate was then transferred within the next 30 min into a thermocycler for a standard amplification procedure. Finally, the amplified droplets were analyzed on a dedicated droplet reader. Therefore, the required ddPCR equipment consisted of 4 devices: (i)A droplet generator for partitioning the PCR mix.(ii)A heat sealer to seal the PCR plate using a special tinfoil sealer to avoid loss of product during the following step, the PCR. The sealer is perforated by the droplet generator upon PCR finalization.(iii)A thermocycler, needed to perform the PCR reaction. In general, a standard PCR device should suffice; however, this protocol was performed on a QX200 Bio-Rad PCR machine. Optimization of conditions may be needed when using different thermocyclers.(iv)A droplet reader (similar to a flow cytometer). It is noteworthy that droplets are stable up to 48 h at 4 °C after the thermocycling reaction.

The actual sensitivity of a ddPCR-CNV assay is dependent on several parameters, including but not limited to the amount of analyzed gDNA. Counterintuitively but importantly, at a certain point, the sensitivity starts to decrease with higher gDNA input. Assessment of the 4-1BB-CAR T reactions relied on positive signals (the FAM fluorophore). When only one target allele was present in a single droplet, the ddPCR should have resulted in either (i) a double positive signal (FAM/HEX); (ii) a FAM-positive HEX-negative signal, both indicating positive 4-1BB-CAR T counts; or (iii) a FAM-negative but HEX-positive signal, representing the housekeeping gene, albumin. In the Bio-Rad system, ~20,000 droplets were generated. The concentration in copies per microliter was thus automatically calculated using the Poisson equation:λ = −ln(1 − p)(1)

Based on the Poisson distribution, ideally ~5000–6000 alleles were analyzed to make sure that the vast majority of droplets contain only a single allele. The upper amount of 6000 alleles corresponded to 3000 human cells and ~20 ng of human gDNA. Thus, using the Bio-Rad systems, the maximum sensitivity was reached at about 20 ng DNA input. Furthermore, as low copy frequency was expected in these experiments, the recommended 20 ng of sample was used per well. 

STEP 6: The ddPCR mastermix is described in Table 1.

STEP 7: Droplet generation was carried out on the Bio-Rad Automated Droplet Generator following the manufacturer’s instructions and ensuring that the correct oil type was selected (probe oil). Immediately following droplet generation, the PCR plate was sealed, and PCR was carried out on a Bio-Rad C1000 Touch Thermal Cycler (STEP 8). The following cycling conditions were used: initial denaturation at 95 °C for 10 min, then 40 cycles of 94 °C for 30 s (ramp 2.5 °C/s) and 58 °C for 1 min (ramp 2.5 °C/s), with a final step of 98 °C for 10 min. Postamplification PCR products were stored at 4–12 °C for a maximum of 48 h before reading.

STEP 9: After thermocycling, plates were transferred to the Bio-Rad QX200 Plate Reader, and the QuantaSoft software was used to run the data collection for ddPCR-CNV with dual FAM and HEX probes following the manufacturer’s instructions. 

### 4.4. ddPCR Analysis

The QuantaSoft software (Bio-Rad) was used for the analysis of these data using the following steps:(1)NTCs were highlighted, and the 2D Amplitude tab (dot-plot) was selected, where a clear distinction between negative and single- or double-positive droplets was made. The middle of the quadrant gate was positioned close to the lower right corner of the negative population. The threshold levels for channels 1 and 2 were set using the quadrant gate option (multiwell thresholding selection).(2)The threshold we routinely used was 600–700 for channel 1 (FAM dye—the 4-1BB-specific probe) and 2000 for channel 2 (HEX dye—the probe for the albumin reference gene).(3)All samples then had the thresholds set as described in step 1.(4)The 4-1BB-specific events were represented in the entire top panel of the quadrant, both as single positives for FAM and as double positives for both FAM and HEX. Single HEX positives were the reference.(5)CNV was calculated.

### 4.5. QuantaSoft Software

Briefly, the QuantaSoft software (QuantaSoft™ Version 1.7, Bio-Rad, London, UK) provides 6 different options in viewing the results: “1D Amplitude”, “2D Amplitude”, “Concentration”, “CNV”, “Ratio”, and “Events”. As discussed previously, the 2D Amplitude (“dot-plot”) was used for analysis, by selecting all wells (Figure 4A) to visualize the data, and for setting the threshold between the two probes. If discrimination among the 4 quadrants (negative, FAM positive, double positive, and HEX positive) was indistinguishable, PCR parameter optimization was carried out. Thresholds were set as previously described to ensure that there was no positive signal on the NTC. On the CNV tab, a plot of the copies for both probes per sample was observed. The indicated “Ratio” was based on the “Concentrations” of events per microliter, which was determined using the total number of “Events”. The Concentration ratio was calculated based on the ratio of the positive droplets of each probe. Then, the CNV was calculated as double the concentration ratio, for a diploid organism, and corrected for Poisson distribution (correction was automatically performed by the software).

### 4.6. Statistics

All statistics in this paper were performed in Excel using the data analysis tool and separate calculations where needed.

## Figures and Tables

**Figure 1 ijms-23-07573-f001:**
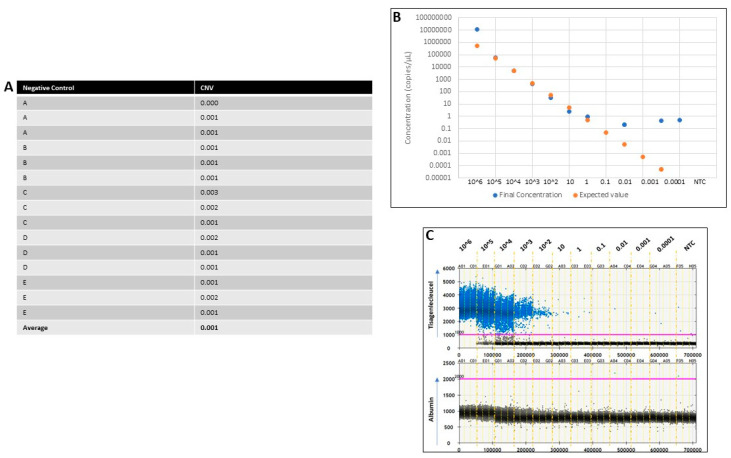
(**A**) Table showing the CNV calculation for each NAC and its replicates (A–E). NACs are nonamplifiable controls and should be negative for CAR Ts but positive for reference genes. The average of all 15 measurements for CAR T cell CNV was 0.001, demonstrating the level of background noise. (**B**) Graph is shown in logarithmic scale. Zero values are not shown when logarithmic scale is used. The final concentration of observed events completely followed expected values between 1 and 10^5^ copies per µL, as expected by the dynamic range of the ddPCR. (**C**) A 1D plot made using the QuantaSoft software (Bio-Rad) showing sequential results of the tisagenlecleucel 4-1BB oligo at copy numbers from 0.0001 to 10^6^. NTCs were also run as negative controls alongside. The blue dots represent the tisagenlecleucel (FAM) and were seen to lose resolution as the copies of the oligo decreased, while the green dots representing albumin (HEX) (the reference gene) and had almost no events, as expected. The 10^4^ dilution was chosen to be used for the next experiment of spiking, as it demonstrated here enough positive events while maintaining a good balance between positive and negative events.

**Figure 2 ijms-23-07573-f002:**
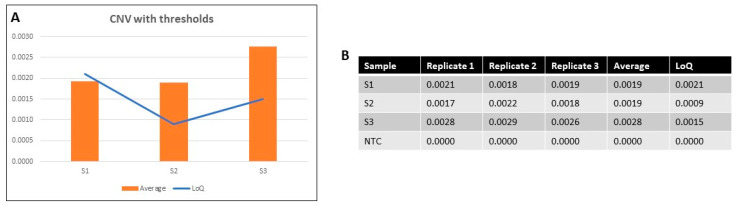
(**A**) Graph demonstrating the CNV average among triplicates per sample for S1, S2, and S3. The blue line shows the LoQ per sample, based on each patient’s pretreatment sample. S2 and S3 showed higher CNV averages than their corresponding LoQs, demonstrating CAR T persistence. (**B**) Table showing the detailed CNV numbers per replicate per sample, their averages, and the corresponding LoQs. NTC was run in parallel as a negative control.

**Figure 3 ijms-23-07573-f003:**
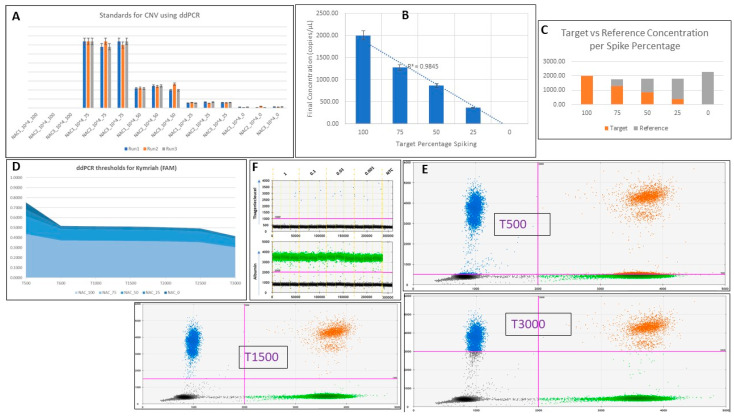
(**A**) Three separate runs with triplicates of three controls (NAC1, NAC2, and NAC3) spiked with the synthetic oligo at proportions of 100% (no control sample, only tisagenlecleucel), 75%, 50%, 25%, and 0% oligo (100% control sample, no tisagenlecleucel). Each replicate was shown to be consistent between runs (*p* = 0.998539). (**B**) Graph demonstrating the correlation between the final concentration (in copies/µL) of the oligo when spiked into the NACs and its corresponding percentage (R^2^ = 0.9845). (**C**) Stacked histogram showing the final concentration in copies/µL of target vs. reference in each triplicate experiment of percentage spiking. (**D**) Area chart showing the accepted range for setting the tisagenlecleucel oligo threshold during analysis. Each NAC spike was run in triplicate for each threshold shown (T500–T3000). (**E**) Representative data for thresholds set at 500 and 1500 for tisagenlecleucel (FAM channel in blue) in 2D amplitude. (**F**) Spikes of the 4-1BB-CAR oligo in control genomic DNA (NACs) at realistic copies down to 0.001. A detailed table with all events can be seen in Table A1. CAR T positive events were detected, but at very low levels. NTC was shown to be zero, as expected.

**Figure 4 ijms-23-07573-f004:**
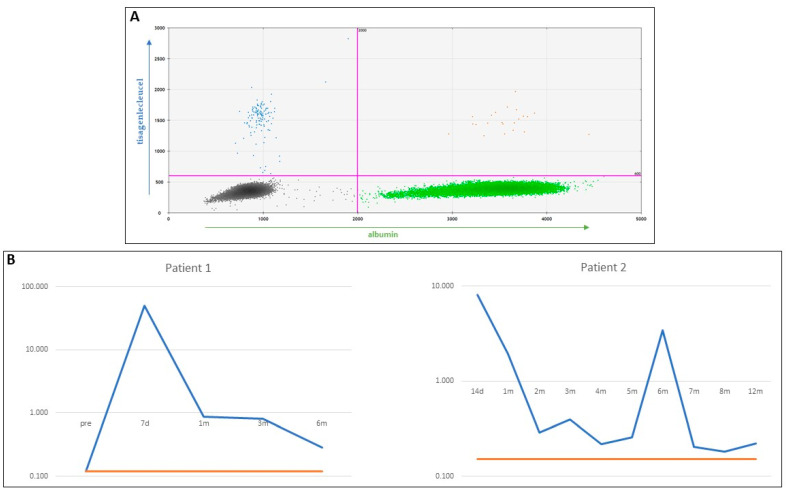
(**A**) A 2D plot showing the 4 quadrants seen in the ddPCR using the QuantaSoft software (Bio-Rad). In the lower left, in black, are the negative events that are always present and in abundance. In the lower right, in green, is the albumin (reference gene). In the upper left, in blue, is the tisagenlecleucel (4-1BB-CAR), and in the upper right, in orange, are the double positive events for both tisagenlecleucel and albumin. The distinction between these populations was clearly visible, allowing for efficient CNV quantification. (**B**) Longitudinal data for patient 1, in log scale, varying from pre-administration of tisagenlecleucel up to 6 months after. The trendline shows the LoQ for this patient’s CAR T levels prior to administration. At 7 days posttreatment, CAR T levels were at the highest level past the LoQ, as expected. The pretreatment sample showed negative CAR T cell levels, also as expected. Also shown are longitudinal data for patient 2, in log scale, from 14 days to a year post-tisagenlecleucel treatment. In this case, CAR T levels never dropped below the LoQ, an indication of good clinical outcome. Raw data can be found in Appendix E.

**Figure 5 ijms-23-07573-f005:**
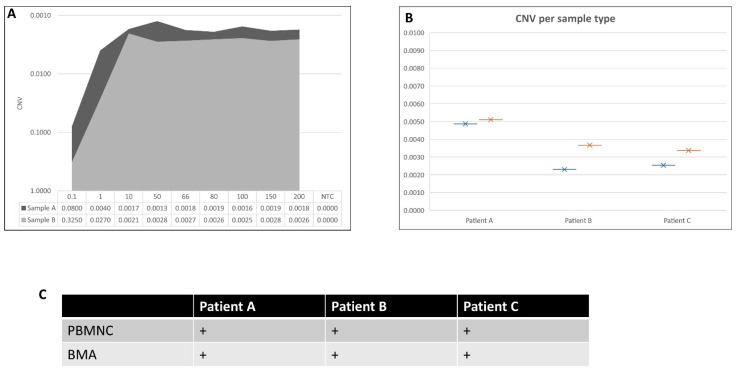
(**A**) Here, two samples, A and B, were used to measure CNV at different concentrations. The area chart demonstrates the CNV, measuring consistency between 10 and 200 ng/µL. A sudden drop in CNV levels was observed below 10 ng/µL for each patient. LoQs were 0.002 and 0.0007 for Samples A and B, respectively. (**B**) Three samples were run here (A, B, and C), with the starting material from each sample being either PBMNC or isolated CD3+ T cells. A comparison between PBMNC samples (blue) and isolated CD3+ T cells (orange) demonstrated consistency between the two different sample types. (**C**) CNV reporting on three patients, with the starting sample material being either PBMNC or BMA (at the same patient timepoint). The table shows when a CNV was reported as above (+), below (−), or equal to (=) the LoQ.

**Figure 6 ijms-23-07573-f006:**
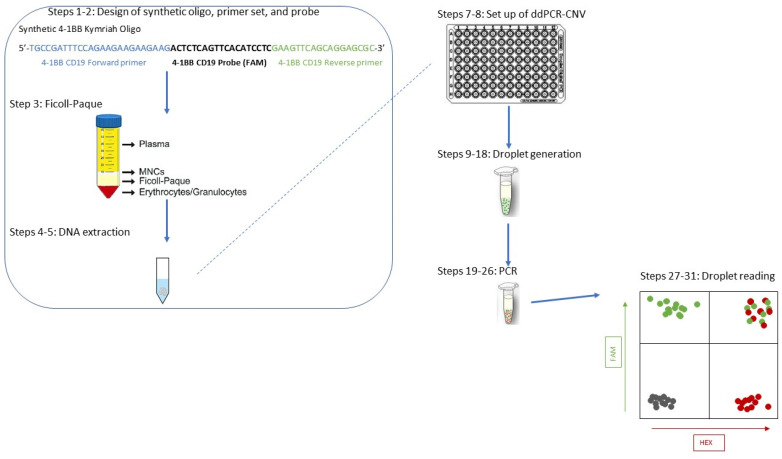
An illustration of a step-by-step procedure for the detection of tisagenlecleucel CAR T cells using the suggested ddPCR-CNV protocol. The design of the oligo and corresponding primers and probes is described in Steps 1–3, followed by the isolation of desired cell population from the patient’s sample taken in clinic (Step 4) and DNA extraction (Step 5). The following steps, 6–9, all relate to the ddPCR-CNV protocol.

**Table 1 ijms-23-07573-t001:** Mastermix for ddPCR.

Reagent	Concentration	Volume per Well (µL)
Supermix	2×	11 µL
F tisagenlecleucel	10 µM	0.8 µL
R tisagenlecleucel	10 µM	0.8 µL
F albumin	10 µM	0.8 µL
R albumin	10 µM	0.8 µL
Probe tisagenlecleucel	10 µM	0.4 µL
Probe albumin	10 µM	0.4 µL
Dra I	20 U/µL	0.1 µL
Sterile water		4.9 µL
Template DNA	10 ng/µL	2 µL

## Data Availability

Not applicable.

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
