# Peer review of "Clinically Applicable Assessment of Tisagenlecleucel CAR T Cell Treatment by Digital Droplet PCR for Copy Number Variant Assessment"

_ijms, 2022, doi:10.3390/ijms23147573_

Round 1

Reviewer 1 Report

In this article the authors have demonstrated a novel digital droplet PCR (ddPCR)-based methodology for the study of CNV 22 in CD19-specific CAR T cells with universal applicability. Some of the specific comments are as follows:

1. The quality of the figures have to be improved.

2. Information regarding statistics has to be included.

3. Page 3 line 126 - page 4 line 139 in the results section is actually talking about the methodology used. It should be moved in the methods section.

4. Results have to be further more and in depth discussed. 

5. Include schematic to better understand the overall purpose/goal of the study.

Author Response

First, we would like to thank you for your comments. We have now addressed all points raised and would like to thank you for your time. We believe your comments have improved our manuscript. Please see our replies in green below:

1. The quality of the figures have to be improved.

Thank you very much for your comments. We have now changed some figures (and relative legends) and improved their resolution.

2. Information regarding statistics has to be included.

This has now been addressed and statistics are included.

3. Page 3 line 126 - page 4 line 139 in the results section is actually talking about the methodology used. It should be moved in the methods section.

This has now been addressed.

4. Results have to be further more and in depth discussed. 

We have now added new figures and text to address this.

5. Include schematic to better understand the overall purpose/goal of the study.

This has now been addressed, please see Appendix A4.

Reviewer 2 Report

In the manuscript titled "Clinically applicable assessment of Tisagenlecleucel CAR T cell treatment by digital droplet PCR for 2 copy number variant assessment" by Soragia Athina Gkazi et al, authors present their efforts to establish a reproducible and efficient method to quantify the proportion of CAR-T cells in patient blood samples in a clinically feasible assay.  Authors present assay development data for a digital droplet PCR approach, and provide several examples of applying the assay to patient samples. The data seems to be sound, the writing of the manuscript is good, and this type of report would likely be of value to the community of researchers working in the area of CAR-T therapies. 

There are a number of points of criticism that would need to be addressed to make this manuscript acceptable for publication:

(1) In figure 1A, authors present artificial data showing enumeration of as few as 0.001 copies/ul of CD19-CAR-T oligonucleotide. The raw event data presented in Figure 1A is fine for exemplary data, but a graph of quantified number versus expected number quantitated over a number of repeats should be presented. Furthermore, authors should provide statistical reasoning for setting the LOD to 0.001, as this limit seems arbitrarily set based on the data presented. 

(2) In figure 1B, authors present a similar experiment wherein human genomic DNA was included to better represent true cellular conditions for quantitation of CAR-T. In this case, authors use a high number of copies of CAR-T oligo to spike into human genomic DNA. This seems like an artificial experiment, as we would never expect to have that number of copies. It would be good to see the same experiment with the ratio of CAR-T to genomic copies titrated down to realistic ratios (0.1, 0.01, 0.001 etc...). 

(3) In figure 1C, authors demonstrate consistency of the experiment by repeatedly quantitating the number of CAR-T copies in patient samples. Authors should spike these samples into non-CAR-T human genomes and confirm that quantitation is accurate down to the LOD. Also a negative control genome should be shown here.

(4) The presentation of the graph in Figure 1C is also poor. What is S1, S2, S3? What is CNV1, 2, 3? A log axis would be more helpful for such an analysis. 

(5) In figure 2, authors present data similar to Fig1B, wherein control human genome (what does NAC stand for), is spiked with high numbers of CAR-T oligos. Again, this experiment seems highly aritificial if authors are mostly intending to apply this to blood samples, where CAR-T copies would be much lower. The data here again is presented poorly. Authors need to improve readability of the figures and explanation in the text. 

(6) In figure 3 authors present longitudinal data for two patients. This data is quite interesting. Is it correct that authors detected low or no CAR-T for most timepoints? Authors should present such graphs in log axis, and show the limit of detection to give better context. The raw data for this should be provided as a supplemental table if possible. 

(7) In Figure 4, authors make a case that that the assay can work with as low as 10ng/ul of human DNA. The way the graphs are presented it is tough to tell whether CAR-T is detected or not. Again, data should be presented in log axis, and negative control/LOD should be shown in parallel. 

(8) The workflow diagram presented in figure 5 is helpful. It would be nice to see a graphical map of the Tisagenlecleucel construct to understand where the primers bind in the sequence. This would help with understanding what other CAR products these specific primers might be used for. 

Overall, the manuscript presents interesting data on an assay that could be valuable to many labs working on CAR-T. Authors must take the time to reformat all figures in the paper to increase readability and improve the data presentation to make the paper acceptable for publication. 

Author Response

First, we would like to thank you for your comments. We have now addressed all points raised and would like to thank you for your time. We believe your comments have improved our manuscript. Please see our replies in green below:

(1) In figure 1A, authors present artificial data showing enumeration of as few as 0.001 copies/ul of CD19-CAR-T oligonucleotide. The raw event data presented in Figure 1A is fine for exemplary data, but a graph of quantified number versus expected number quantitated over a number of repeats should be presented. Furthermore, authors should provide statistical reasoning for setting the LOD to 0.001, as this limit seems arbitrarily set based on the data presented. 

Thank you very much for your comments. We have now extensively amended these figures and have included a new graph showing observed vs expected data and an explanation on LoD setting as requested.

(2) In figure 1B, authors present a similar experiment wherein human genomic DNA was included to better represent true cellular conditions for quantitation of CAR-T. In this case, authors use a high number of copies of CAR-T oligo to spike into human genomic DNA. This seems like an artificial experiment, as we would never expect to have that number of copies. It would be good to see the same experiment with the ratio of CAR-T to genomic copies titrated down to realistic ratios (0.1, 0.01, 0.001 etc...). 

This is an interesting point. We have performed a new experiment and included the data in the manuscript as suggested.

(3) In figure 1C, authors demonstrate consistency of the experiment by repeatedly quantitating the number of CAR-T copies in patient samples. Authors should spike these samples into non-CAR-T human genomes and confirm that quantitation is accurate down to the LOD. Also a negative control genome should be shown here.

Unfortunately, we cannot perform these experiments because we receive very low volume samples from post-Kymriah paediatric patients, who also have low cell counts. Thus, we do not have sufficient DNA to carry out multiple runs from these samples.

However, we attempt to address this as best we can by comparing each patient’s post-Kymriah samples to their pre-Kymriah samples.

(4) The presentation of the graph in Figure 1C is also poor. What is S1, S2, S3? What is CNV1, 2, 3? A log axis would be more helpful for such an analysis. 

Apologies for the poor presentation. We have now amended the resolution of the new figures and their legends for better clarity.

(5) In figure 2, authors present data similar to Fig1B, wherein control human genome (what does NAC stand for), is spiked with high numbers of CAR-T oligos. Again, this experiment seems highly aritificial if authors are mostly intending to apply this to blood samples, where CAR-T copies would be much lower. The data here again is presented poorly. Authors need to improve readability of the figures and explanation in the text. 

As part of the new amendments of the paper, this should now be addressed.

(6) In figure 3 authors present longitudinal data for two patients. This data is quite interesting. Is it correct that authors detected low or no CAR-T for most timepoints? Authors should present such graphs in log axis, and show the limit of detection to give better context. The raw data for this should be provided as a supplemental table if possible. 

We have now added a new figure here with thresholds added on a log scale. Apologies for the misunderstanding, usually samples are found to be above threshold. We have provided the raw data.

(7) In Figure 4, authors make a case that that the assay can work with as low as 10ng/ul of human DNA. The way the graphs are presented it is tough to tell whether CAR-T is detected or not. Again, data should be presented in log axis, and negative control/LOD should be shown in parallel. 

This has now been addressed.

(8) The workflow diagram presented in figure 5 is helpful. It would be nice to see a graphical map of the Tisagenlecleucel construct to understand where the primers bind in the sequence. This would help with understanding what other CAR products these specific primers might be used for. 

We have added a supplementary figure to address this, although we do not have the full DNA sequence from Novartis for Kymriah since this is their legal entity.

Reviewer 3 Report

The manuscript entitled: “Clinically applicable assessment of Tisagenlecleucel CAR T cell
treatment by digital droplet PCR for copy number variant assessment (ijms-1757566)” by Soragia Athina Gkazi et al. investigates the novel digital droplet PCR based methodology for the study of CNV in 4-1BB CD19-specific CAR T cells.

Albeit the paper is well written, prepared and of special interest, some comments should be addressed.

Comments:

1.     Result section line 127-128. Please add the reference.

2.     Discussion section: Line 193-194: Are the results all over within less than one day available? Please clarify “inexpensive” in more detail.

3.     Discussion section: The authors should pronounce in more detail the strengths and weaknesses in clinical practice. Moreover, the authors should mention if there is need for further re-evaluation?

4.     Figure 1-4 are too small and need to be enlarged.

Author Response

First, we would like to thank you for your comments. We have now addressed all points raised and would like to thank you for your time. We believe your comments have improved our manuscript. Please see our replies in green below:

  1. Result section line 127-128. Please add the reference.

This has now been added.

  1. Discussion section: Line 193-194: Are the results all over within less than one day available? Please clarify “inexpensive” in more detail.

This has now been addressed and we have provided sample costing.

  1. Discussion section: The authors should pronounce in more detail the strengths and weaknesses in clinical practice. Moreover, the authors should mention if there is need for further re-evaluation?

We have now added a new figure (Appendix A4) to show where the study fits into clinical practice. We have also added extra figures in Results showing the strengths and weaknesses of the study, including more detail for the LoD and LoQ.

  1. Figure 1-4 are too small and need to be enlarged.

All figures have been changed in resolution and a few of them have now changed along with their legends.

Round 2

Reviewer 2 Report

Thank you for your revisions. The changes to the manuscript have addressed many of my concerns, though there remains a few areas of weakness:

(1) All of the figures in the paper are quite blurry and hard to read. It is not possible to make out individual dots in many of the ddPCR plots. This may be an artifact of the manuscript review draft process, but this would absolutely need to be addressed. Authors must address this issue by providing higher graphical quality figures. All text must be readable and datapoints visible. 

(2) It would be helpful to see a chart or table presenting the expected versus observed number of positive events for the data presented in Figure 1D. It is important to have a robust assessment of the consistency in this experiment in particular as it is the most well controlled and realistic assessment of CAR-T quantitation. 

(3) Data presented in Figure 2 is improved. Figure legend should make it clear that these samples are pre-treatment and would be expected to be negative. 

(4) Figure 3 is a very important experiment but without decreasing the spike levels closer to CNV numbers that are more realisitic, it is not helpful to determine the accuracy of this test. The lowest concentration tested is 25% spike-in (0.25), whereas authors claim a LOD of 0.001 and present detection of patient values in the range of 0.002 in Figure 2A. The data presented in Figure 3 must be repeated with spike in levels going down to 0.001, and also should include a negative control (no spike).

(5) The revised log plots presented in Figure 4 are much improved. The raw data provided in Appendix A5 is helpful. 

(6) The revised figure 5 is improved, but needs further revision. It is not clear why an inverted log axis is used in Fig 5a. This should be a positive log axis, and the expected value should be shown on the graph. This will make it clear that the results are diverging from expectation at <10ng/uL

(7) Line 227 - References to former Fig 4 need to be updated. 

(8) Figure 6 is not referenced in the manuscript, but should be. This could be referred to in the discussion section. 

(9) Throughout the manuscript, authors refer to 4-1BB CD19 CAR, but this is a bit confusing. It would be fair to say that the 4-1BB transgene is detected. At first definition (line 59) it could be referred to as detection of "the tisagenlecleucel 4-1BB signaling domain (41BB-CAR)". 4-1BB-CAR could then be used throughout the rest of the manuscript, as strictly speaking this is only detecting any DNA from the transgenic 4-1BB domain and not specific to CD19. 

(10) Image provided in Appendix A3 appears to be taken from another publication. Authors should generate their own version of this figure if they wish to include it in the manuscript appendix or remove the figure altogether. 

Overall, authors are encouraged to further improve the quality of this submission to bring it to the level that would be appropriate for publication. 

Author Response

Thank you very much for your comments. Please see our replies highlighted in green:

(1) All of the figures in the paper are quite blurry and hard to read. It is not possible to make out individual dots in many of the ddPCR plots. This may be an artifact of the manuscript review draft process, but this would absolutely need to be addressed. Authors must address this issue by providing higher graphical quality figures. All text must be readable and datapoints visible. 

Thank you for your comments. It must be a review draft artifact as you suggested. We can assure you that all figures have been saved in 300 dpi resolution and sent separately to the journal as part of the re-submission.

(2) It would be helpful to see a chart or table presenting the expected versus observed number of positive events for the data presented in Figure 1D. It is important to have a robust assessment of the consistency in this experiment in particular as it is the most well controlled and realistic assessment of CAR-T quantitation. 

Figure 1D has now been moved to Figure 3F as part of the spiking dynamics. It is also now accompanied by a table of all observed events per spiking levels in Appendix A1.

(3) Data presented in Figure 2 is improved. Figure legend should make it clear that these samples are pre-treatment and would be expected to be negative. 

This has now been added to the legend.

(4) Figure 3 is a very important experiment but without decreasing the spike levels closer to CNV numbers that are more realisitic, it is not helpful to determine the accuracy of this test. The lowest concentration tested is 25% spike-in (0.25), whereas authors claim a LOD of 0.001 and present detection of patient values in the range of 0.002 in Figure 2A. The data presented in Figure 3 must be repeated with spike in levels going down to 0.001, and also should include a negative control (no spike).

This has now been amended and an events table has been added for further detail in Appendix A1.

(5) The revised log plots presented in Figure 4 are much improved. The raw data provided in Appendix A5 is helpful. 

Thank you for your comment. We are happy that these changes have added clarity.

(6) The revised figure 5 is improved, but needs further revision. It is not clear why an inverted log axis is used in Fig 5a. This should be a positive log axis, and the expected value should be shown on the graph. This will make it clear that the results are diverging from expectation at <10ng/uL

The log axis has been revised and is now not inverted, thank you for noticing this. LoQs for both samples have also been added to the legend.

(7) Line 227 - References to former Fig 4 need to be updated. 

This has now been updated.

(8) Figure 6 is not referenced in the manuscript, but should be. This could be referred to in the discussion section. 

This has now been added.

(9) Throughout the manuscript, authors refer to 4-1BB CD19 CAR, but this is a bit confusing. It would be fair to say that the 4-1BB transgene is detected. At first definition (line 59) it could be referred to as detection of "the tisagenlecleucel 4-1BB signaling domain (41BB-CAR)". 4-1BB-CAR could then be used throughout the rest of the manuscript, as strictly speaking this is only detecting any DNA from the transgenic 4-1BB domain and not specific to CD19. 

This has now been amended as suggested.

(10) Image provided in Appendix A3 appears to be taken from another publication. Authors should generate their own version of this figure if they wish to include it in the manuscript appendix or remove the figure altogether. 

This has now been removed.

Round 3

Reviewer 2 Report

Authors have addressed much of my concerns, but given the quality of the figures I cannot yet endorse this for publication. I will request better quality figures from the editor. 

Round 4

Reviewer 2 Report

Authors have adequately addressed my concerns. Report as written is now acceptable for publication.